# Advancing *Cordyceps militaris* Industry: Gene Manipulation and Sustainable Biotechnological Strategies

**DOI:** 10.3390/bioengineering11080783

**Published:** 2024-08-02

**Authors:** Yan Hu, Yijian Wu, Jiayi Song, Maomao Ma, Yunzhu Xiao, Bin Zeng

**Affiliations:** College of Pharmacy, Shenzhen Technology University, Shenzhen 518118, China; hbqinghe0806@163.com (Y.H.); wuyijian0101@163.com (Y.W.); 2201459@stu.neu.edu.cn (J.S.); ma_maomao@hotmail.com (M.M.); xiaoyunzhu8891@163.com (Y.X.)

**Keywords:** *Cordyceps militaris*, gene editing, synthetic biology, sustainable production

## Abstract

*Cordyceps militaris* is considered to be of great medicinal potential due to its remarkable pharmacological effects, safety, and edible characteristics. With the completion of the genome sequence and the advancement of efficient gene-editing technologies, coupled with the identification of gene functions in *Cordyceps militaris*, this fungus is poised to emerge as an outstanding strain for medicinal engineering applications. This review focuses on the development and application of genomic editing techniques, including *Agrobacterium tumefaciens*-mediated transformation (ATMT), PEG-mediated protoplast transformation (PMT), and CRISPR/Cas9. Through the application of these techniques, researchers can engineer the biosynthetic pathways of valuable secondary metabolites to boost yields; such metabolites include cordycepin, polysaccharides, and ergothioneine. Furthermore, by identifying and modifying genes that influence the growth, disease resistance, and tolerance to environmental stress in *Cordyceps militaris*, it is possible to stimulate growth, enhance desirable traits, and increase resilience to unfavorable conditions. Finally, the green sustainable industrial development of *C. militaris* using agricultural waste to produce high-value-added products and the future research directions of *C. militaris* were discussed. This review will provide future directions for the large-scale production of bioactive ingredients, molecular breeding, and sustainable development of *C. militaris*.

## 1. Introduction

*Cordyceps militaris*, a well-known edible and medicinal fungus, is considered to be a valuable mushroom used in traditional Chinese medicines and health supplements, which was approved by China’s Ministry of Health as a new resource of food in 2009. *C. militaris* primarily parasitize the pupae and larvae of Lepidoptera insects, which can penetrate the host’s body via the cuticle, spiracles, mouthparts, or other access points. The infection process of Cordyceps involves three stages: invasion, parasitism (development of the fungus before the death of the insect), and saprophytism (growth of the fungus after the host’s death) [1]. *C. militaris* has many bioactive constituents, like cordycepin, polysaccharide, cordyceps acid, and ergosterol, as well as multiple microelements [2,3], among which cordycepin (3′-deoxyadenosine), an adenosine analog, is one of the most valuable and extensively studied components; it boasts various pharmacological functions: antioxidant, anticancer, antitumor, antibacterial, and anti-inflammatory activity, and immune regulation [4,5,6,7,8]. Moreover, compared to *Cordyceps sinensis*, it grows more rapidly, requires less stringent cultivation conditions, and some of its key bioactive ingredients are even higher than *Cordyceps sinensis*, so *C. militaris* is widely used as an ideal substitute for *C. sinensis* and is of great research value. Traditionally, the most direct method for obtaining bioactive secondary metabolites like cordycepin from *C. militaris* is through extraction from wild or artificially cultivated fruit bodies [9,10]. However, due to the long growth cycle and strict environmental requirements, this process incurs high technical costs and also adds to the environmental burden of the industry.

In recent years, with the completion of the *C. militaris* whole-genome sequencing [11], research in molecular biology and genetics has been progressively deepening. Through genetic engineering, researchers can precisely edit the genome of *C. militaris* to enhance its ability to produce specific bioactive components, such as cordycepin [12,13]. In particular, CRISPR/Cas9 gene-editing technology enables the modification of multiple genes in both functional genomics research and strain breeding in *C. militaris* [14,15,16]. In addition, the construction of a high-production strain for secondary metabolites can be achieved by metabolic engineering and homologous/heterologous expression [17,18,19]. These technological advancements not only help to improve the biosynthetic efficiency of *C. militaris* but may also reveal its complex bioactive mechanisms, laying the foundation for the development of new drugs to treat various diseases.

Although *C. militaris* can be commercially cultivated on a large scale, it suffers from degradation problems similar to other edible fungi such as *Flammulina velutipes*, Lentinula edodes, and *Pleurotus eryngii* [20,21,22]. To date, the challenge of *C. militaris* degeneration remains unresolved, and its molecular mechanisms are not well understood [23,24]. Additionally, fungal diseases have become one of the bottlenecks in the scaled production of fruiting bodies, with “white mildew” disease caused by *Calcarisporium cordycipiticola* being the most damaging [25,26]. This not only leads to reduced yields of *C. militaris* but also raises concerns about pesticide contamination and product safety. Fortunately, the establishment of the first disease-resistant gene-editing breeding system based on CRISPR/Cas9 technology in *C. militaris* represents a seminal advance [27]. This development underscores the potential of gene-edited, disease-resistant cultivars as a cost-effective and environmentally sustainable strategy to significantly reduce losses attributed to agricultural diseases. Here, we present a review of the application of gene-editing techniques to improve the growth and metabolite production in *C. militaris*, and the potential of synthetic biology to optimize metabolic pathways and increase yield was discussed, alongside innovative approaches to combat industry challenges such as degradation and disease. Finally, the existing challenges and future research direction of *C. militaris* were discussed, aiming to promote the scientific development and technological innovation of the *C. militaris* industry.

## 2. The Genetic Manipulation Technology of *Cordyceps militaris*

### 2.1. Commonly Used and Efficient Genetic Transformation System

#### 2.1.1. *Agrobacterium tumefaciens*-Mediated Transformation

The *Agrobacterium tumefaciens*-mediated transformation system for *C. militaris* was successfully established and optimized in 2011 [28], when *Agrobacterium* AGL-1 harboring the vector pATMT1 exhibited a transformation efficiency of 30–600 transformants per 1 × 10^5^ conidia. Based on this, researchers utilized ATMT technology to create a mutant library of *C. militaris* and screened mutants using degenerate characteristics [29]. The crucial advantage of the ATMT method is the flexibility in choosing recipient cell types as starting materials for transformation. In particular, fungal spores serve as the most common material because millions of spores can be easily collected from a culture plate [30]. However, some challenges, including complicated and laborious procedures, low rates of homologous recombination, as well as a lot of transformant screening work, should be taken into consideration [31].

#### 2.1.2. Polyethylene Glycol (PEG)-Mediated Protoplast Transformation

PEG is a kind of polymer with stable physicochemical properties, which can promote the fusion of exogenous DNA and protoplasts by interfering with the surface charge of the cell membrane or acting as an adhesive between DNA and cell membrane. This method was first successful in *Saccharomyces cerevisiae* [32] and was later applied to filamentous fungi. The preparation of protoplasts of *C. militaris* was discussed in several essays [14,16,33,34,35]. The *Tns* gene, which encodes a terpene synthase in *C. militaris*, was successfully knocked out using a combination of split-marker methods and PEG-mediated protoplast transformation techniques [33]. Moreover, the efficiency of PEG-mediated protoplast transformation in *C. militaris* was improved by adding the surfactant Triton X-100 used to enhance cell membrane permeability [15]. In addition, PEG-mediated protoplast transformation was markedly more efficient than ATMT transformation; the same plasmid was transformed by PEG-mediated protoplast transformation and ATMT; among the 38 transformants obtained by PEG-mediated protoplast transformation, 35 showed the predictable phenotype; however, only 7 were observed among the 70 transformants obtained by ATMT [16].

### 2.2. Selective Marker for the Detection of Positive Transformants

A stable and efficient genetic transformation system requires suitable screening markers, which can reduce false positives, improve screening efficiency, and avoid the loss of foreign genes in the process of purification and culture. Commonly used screening makers include auxotrophic markers and drug-resistant makers.

#### 2.2.1. Auxotrophic Markers

An auxotrophic mutant strain refers to a mutant strain that loses the ability to synthesize a certain growth factor and can only grow in a medium with an exogenous addition of the growth factors. Auxotrophic screening restores the wildtype growth of recipient cells by transferring auxotrophic complementary genes into recipient cells. The auxotrophic markers currently used in *C. militaris* are *pyrG* (encoding orotic acid-5′-monophosphate decarboxylase) and *hisB* (encoding imidazole glycerophosphate dehydratase) [14,36]. So, we screen the positive transformants by adding uridine/uracil or histidine. Recently, the Δ*pyrG* Δ*hisB* dual auxotrophic mutant transformation system was constructed, and the knockout and complementation of the photoreceptor gene Cm*WC*-1 were accomplished [36]. The newly constructed ATMT system with histidine and uridine/uracil auxotrophic markers provides a promising tool for genetic modifications in the medicinal fungus *C. militaris.*

#### 2.2.2. Drug-Resistance Markers

Resistance screening refers to the transfer of resistance genes into recipient cells to make the transformants produce drug resistance, which can be screened at a certain drug concentration. Hygromycin, G418, phleomycin, nourseothricin, Basta, 5-fluorocytosine, and 5-FOA are commonly used in the genetic manipulation of fungi. The growth of conidia and mycelia was completely inhibited at a Basta concentration of 0.4 g/L and a 5-FOA concentration of 0.1 g/L, respectively [14]. After other resistant plasmids were transferred into recipient cells, related resistance was added to the plate for screening [33,37,38].

### 2.3. Split-Marker Approach

The split-marker approach was established to improve the frequency of homologous integration and gene knockout [33]. Deletion cassettes with different forms of the selectable marker (split or linear) have been made using single-joint PCR (SJ-PCR) and double-joint PCR (DJ-PCR). Then, the linear and split-marker deletion cassettes were constructed and introduced into *C. militaris* protoplasts by PEG-mediated transformation. The gene-targeting specificity and precision were enhanced when using split-marker fragments compared to linear deletion cassettes. The transformation of split-marker fragments led to a greater efficacy in achieving targeted gene disruption compared to the use of linear deletion cassettes. Moreover, the mutants could maintain resistance to glufosinate ammonium. While this approach has significantly increased the frequency of homologous recombination, the restricted selection marker choices and the enduring presence of resistance pose significant challenges in subsequent genome editing [39].

### 2.4. CRISPR-Cas9 Gene-Editing Technology

The clustered regularly interspaced short palindromic repeats (CRISPR) and CRISPR-associated protein 9 (Cas9) system represents a major breakthrough in the field of biology and medicine in recent years, providing an efficient and flexible method for accurately editing genes. The CRISPR editing system was first established in *C. militaris* in 2018, the *pyrG* gene was successfully edited by expressing the codon-optimized Cas9 gene with the newly discovered promoter Pcmlsm3 and terminator Tcmura3; however, the efficiency was only 11.76% [14]. The editing efficiency of *Cmura*5 reached 100% when an optimized ribonucleoprotein (RNP)-based method was employed [15]. Subsequently, marker-free multiplex gene-editing and a large DNA fragment deletion technique were developed by mining endogenous tRNA processing elements, introducing the autonomously replicating AMA1 plasmid and homologous templates [40]. Additionally, using the AMA1-based autonomously replicating CRISPR-Cas9 gene-editing system, the blue light receptor genes WC-1 and VVD in *C. militaris* were edited with efficiencies of 55.1% and 89%, respectively [16]. The application of CRISPR-Cas9 technology has accelerated the development of molecular breeding in *C. militaris* and improved the production efficiency of bioactive fungal strains. The gene transformation method of *C. militaris* is shown in Table 1 and Figure 1.

## 3. Increasing the Content of Secondary Metabolites of *C. militaris* by Genetic Engineering

*C. militaris* contains various active secondary metabolites, like cordycepin (COR), cordyceps polysaccharides, ergothioneine, carotenoid, and other active substances. With the publication of the whole-genome sequence of *C. militaris* and the development of genomics, metabolomics, transcriptomics, and other methods and technologies, the biosynthetic pathways of these metabolites have been gradually illustrated, which is conducive to increasing the content of secondary metabolites of *C. militaris* by genetic engineering.

### 3.1. The Enhancement of COR Production

To create industrial cell factories that produce high amounts of cordycepin, it is necessary to investigate the gene cluster responsible for cordycepin synthesis. Despite the identification of cordycepin in *C. militaris* in 1950 [42], the deciphering of its biosynthetic pathway has been protracted due to the absence of sophisticated genome analysis techniques. It was not until 2011 that the adenosine metabolic pathways in *C. militaris* and the genes involved in cordycepin synthesis were predominantly uncovered through genomics and transcriptomics studies. In 2017, Xia’s research team identified four highly conserved protein-coding genes by evaluating a vast number of orthologous proteins between *C. militaris* and *Aspergillus nidulans*, designated as cns1-cns4 (CCM_04436-CCM_04439), which were associated with adenosine metabolism in *C. militaris* [43]. The latest insights into the cordycepin biosynthesis pathways have been compiled in Figure 2. The results showed that the initial precursor of cordycepin biosynthesis pathway was glucose [44,45]. Glucose is transported into the interior of *C*. *militaris*, where it is phosphorylated to form glucose-6-phosphate (G6P), which is diverted into two pathways: a portion enters the pentose phosphate pathway, while the remainder is converted to fructose-6-phosphate (F6P) by phosphoglucose isomerase. F6P subsequently enters the tricarboxylic acid cycle (TCA) to provide energy for the growth of cells. The G6P that enters the pentose phosphate pathway is transformed into ribose-5-phosphate, which is converted into inosine monophosphate (IMP) through a series of reactions. IMP is then converted into adenylosuccinate and transformed into cordycepin under the catalysis of the cordycepin substrate channel.

Zhang et al. investigated the function of ribonucleotide reductases (RNRs), including the two subunits RNRL and RNRM, in the production of cordycepin (COR) by enhancing the expression of these genes in *C. militaris* [13]. The results showed that the concentration of cordycepin (COR) was significantly increased in *C. militaris* cells where the RNRM gene was overexpressed, reaching 3.750 mg/g. In contrast, the COR content in the cells overexpressing the RNRL gene only saw a minor increase to 2.600 mg/g, which was not significantly different from the wildtype level of 2.500 mg/g. It was suggested that RNRM might directly contribute to the biosynthesis of COR by hydrolyzing adenosine, although the precise mechanism by which RNRM facilitates COR synthesis remains to be elucidated. Research indicates that low-oxygen environments favor the synthesis and secretion of cordycepin, and the hypoxic environment within the insect hemocoel may also contribute to the higher cordycepin content in fruiting bodies growing on pupae [46]. Improving the low-oxygen environment by overexpressing sterol regulatory element-binding proteins (SREBPs) significantly increases cordycepin content [47]. Overexpressing enzymes involved in cordycepin biosynthesis, including the adenylosuccinate synthase, adenylosuccinate lyase, and 5′-nucleotidase genes, led to increased production in recombinant strains compared to the wildtype. Among 24 recombinant strains, the CM-adss-5 strain had the highest cordycepin production, and the extracellular and intracellular cordycepin contents were 1119.75 ± 1.61 and 65.56 ± 0.97 mg/L, 1.26 and 2.61 times those of *C. militaris* WT [48]. In addition, cordycepin can also be produced by heterologous expression of the key genes of cordycepin synthesis, such as overexpression of Cns1 and Cns2 in *Aspergillus oryzae*; through optimized fermentation conditions, a cordycepin yield of 564.64 ± 9.59 mg/L/d has been achieved [49]. As a result of overproduction of cordycepin in *Saccharomyces cerevisiae* by cordycepin synthase screening and metabolic engineering, the COR titer reached 725.16 mg L^−1^ in a 5 L bioreactor [50]. During high-level de novo biosynthesis of cordycepin by systems metabolic engineering in *Yarrowia lipolytica*, the production of cordycepin was achieved at 3588.59 mg/L from glucose [51,52]. By elucidating the biosynthesis and regulatory mechanisms of COR, genetically modifying the genes related to COR biosynthesis in *C. militaris* will become a viable approach to enhance the production of COR in this organism.

### 3.2. The Enhancement of Cordyceps Polysaccharides Content

The activity of polysaccharides is determined by their composition of monosaccharides, the type of glycosidic links, and the level of polymerization [53]. Cordyceps polysaccharides are mainly composed of glucans connected by glycosidic bonds, which include (1 → 3)- and (1 → 6)-β or (1 → 3)-, (1 → 4)-, and (1 → 6)-α-glucans [54,55]. Investigations into the biosynthesis of polysaccharides in various organisms, such as the overexpression of crucial polysaccharide biosynthetic genes [56], the co-expression of multiple genes [57], and the inhibition of polysaccharide synthesis pathways [58], represent all viable strategies to enhance yield. The proposed biosynthetic pathway of polysaccharides from *C. militaris* was shown in Figure 3.

The pathway of fungal polysaccharide biosynthesis encompasses three stages: the synthesis of nucleotide sugar precursors, the arrangement of these precursors into repeating units, and the subsequent polymerization of these units [59]. Therefore, enhancing the catalytic efficiency of these enzymes through overexpression is anticipated to boost the yield of polysaccharides. Researchers overexpressed four genes—gk (glucokinase), pgm (phosphoglucomutase), ugp (UDP-glucose-pyrophosphosphprylase), and upgh (UDP-glucose-6-dehydrogenase)—in *C. militaris*, with the mutant CM-pgm producing 4.63 ± 0.23 g/L of polysaccharide, a 34.89% increase over the wildtype’s 3.43 ± 0.26 g/L. Furthermore, the combined overexpression of pgm and upgh in the engineered strain led to a polysaccharide yield of 6.11 ± 0.21 g/L, a 78.13% improvement over the wildtype. The study showed that pgm is an important regulatory gene that affects EPS biosynthesis of *C. militaris* [37]. Production of polysaccharides in *C. militaris* is intricate and multifaceted, involving a wide array of structural forms and gene clusters. Additional investigation is required to delve into the mechanisms governing polysaccharide biosynthesis and its regulation.

### 3.3. The Enhancement of Other Active Secondary Metabolites Production

Ergothioneine, one of the ingredients with great antioxidant and anti-inflammatory activity [60] in *C. militaris*, is a natural sulfur-containing thiol molecule, and has been regarded as longevity vitamins these years [61,62]. However, ergothioneine could not be synthesized by plants and animals, including humans, and the primary dietary source of ergothioneine for humans is mushroom-based foods [63]. To enhance the commercial yield of ergothioneine, engineered strains featuring different ergothioneine synthetases have been developed in model organisms such as *Escherichia coli* [64,65,66], *Saccharomyces cerevisiae* [67,68], *Aspergillus oryzae* [69], etc. [70,71]. However, despite the intricate optimization and high costs associated with fermentation, the ergothioneine content is not high enough to fulfill the market’s needs. The ergothioneine content in the wildtype *C. militaris* fruiting body ranged from 382 to 799 mg/kg of dry weight (DW), and 130 ± 11 mg/kg DW mycelia. Chen et al. identified ergothioneine synthetase genes in mushrooms and engineered them into *C. militaris* to establish a novel ergothioneine biosynthetic route. By integrating this route into the *C. militaris* genome, they achieved a significant boost in the production of ergothioneine, with the engineered strain producing up to 2.5 g/kg dry weight [72].

Carotenoids, including lutein, zeaxanthin, and cordyxanthin, are a class of pigments that possess a common terpenoid backbone, which can be used as food, dietary supplements, and cosmetic additives [73,74]. Although the biosynthetic pathway of *C. militaris* carotenoids is still unknown, some genes have been identified to be closely related to carotenoid synthesis. The color of *C. militaris* colonies can transform from white to yellow or orange when exposed to light, a phenomenon attributed to the accumulation of carotenoids. WC-1 is an essential component of all known blue light responses [75,76,77]. The absence of the CmWC-1 gene leads to abnormal color change and impaired synthesis of carotenoids [78]. Similarly, the deletion of the photoreceptor gene VVD also impaired the synthesis of carotenoids [79]. By analyzing transcriptome data of *C. militaris* 10 cultured under dark and light conditions, the CCM_06728 and CCM_09155 genes might be involved in the biosynthesis of carotenoids in *C. militaris*, and the absence of terpenoid synthase gene (Tns) in *C. militaris* led to a significant decrease in carotenoid production [80]. To obtain higher yield of carotenoid, it is urgent to study the biosynthetic pathway and related genes of *C. militaris* carotenoids in order to improve the content of carotenoids by genetic engineering.

Pentostatin is a potent inhibitor of adenosine deaminase (ADA) [81], which was approved by the FDA in 1991 and became a commercial drug for the treatment of hairy cell leukemia (under the brand name Nipent). During the majority of the growth cycle of *C*. *militaris*, pentostatin functions intracellularly by inhibiting the activity of adenosine deaminase, ensuring the synthesis of cordycepin and its low abundance presence within the cell [43]. Pentostatin shares the same precursor with cordycepin [43]; its production can be enhanced by deleting *cns*1 [82]. With the discovery of biosynthetic pathways of active ingredients in *C. militaris*, overexpression of key synthetic genes or knockout genes of competing pathways by genetic engineering technology is expected to increase the yield of active substances.

## 4. Promoting the Growth and Development of *C. militaris* by Genetic Engineering

The cultivation of *C. militaris* involves four key developmental stages: mycelial culture, color induction, primordia simulation, and fruiting body formation [29,83]. Using genetic engineering technology to regulate the growth and development genes of *C. militaris*, improve the growth rate, shorten the growth cycle, and reduce the culture cost will be conducive to the development of this industry. Recently, researchers have made significant progress in the identification and functional exploration of genes related to the traits of *C. militaris* (Table 2).

Hydrophobins are a unique class of small molecular secretory proteins in fungi and play a crucial role in their growth and development. *C. militaris* includes four hydrophobins. CmHYD1 and CmHYD2 are class II hydrophobins, while CmHYD3 and CmHYD4 are class I hydrophobins. Gene knockout, complementation, and overexpression studies have revealed that CmHYD1 positively regulates mycelial growth, hydrophobicity, and resistance to osmotic and hydrogen peroxide stress, as well as fruiting-body development, which regulates conidiospore formation by modulating the transcription levels of the circadian clock protein gene Cmfrq and the velvet factor gene Cmvosa [84]. In contrast, the knockout of the CmHYD4 gene promotes conidiospore formation and increases the density of fruiting bodies, enhancing the content of carotenoids and adenosine [85]. The first identified hydrophobin was *SC*3 from the *Schizophyllum commune*, which is essential for the formation of aerial mycelium [86]. Reports on hydrophobins in macrofungi also exist, for instance, nine hydrophobin genes were identified in *Tricholoma vaccinum*, which play an important role in the growth of aerial hyphae and fruiting-body development [87]. In *Flammulina filiformis*, *Hyd*9 was confirmed to play a significant role in the aerial hyphae and fruiting body formation through RNAi and overexpression [88].

Lectins are a class of proteins or glycoproteins that are not derived from the immune system and are widely found in plants, animals, and fungi [89]. Research has confirmed that lectins are closely related to the growth and development of macrofungi. For example, the application of *Agrocybe aegerita* lectin on the surface of normally growing mycelium can significantly promote the differentiation of mycelium and the formation of primordia in *Agrocybe aegerita* and *Auricularia polytricha* [90]. Additionally, the activity of *Grifola frondosa* lectin is highest in fruiting bodies and lower in mycelium and primordia [91]. Four types of lectins have been identified in *C. militaris* [38,92,93,94], among which the CmLec3 gene is expressed at a level five times higher in primordia than in mycelium and 1.35 times higher in fruiting bodies than in mycelium, which may be associated with the development of fruiting bodies [94]. CmLec4 may regulate various functions within *C. militaris* by interacting with polysaccharides and is involved in host invasion and the formation of fruiting bodies [38]. Compared to plants and animals, research on lectins in large fungi is relatively lagging, and further studies on the mechanisms of action of lectins at the genetic level and how they promote fungal development are needed.

Light is also a crucial environmental factor that significantly affects the growth, development, and metabolism of *C. militaris*. Seven types of photoreceptors have been identified in *C. militaris*, including white collar (WC-1), WC-2, Drosophila–Arabidopsis–Synechocystis–human-type cryptochromes (CRY-DASH), CRY-2, cyclobutane pyrimidine dimer photolyase (CPD), VIVID (VVD), and phytochrome (PHY) [11]. The deletion of Cry-DASH in *C. militaris* results in reduced conidial numbers but significantly increased levels of carotenoids and cordycepin, and the light-induced expression of Cmcry-DASH is dependent on the expression of Cmwc-1 [95]. Studies suggest that CRY-DASH can repair damaged single-stranded and double-stranded DNA and its bound CPD, but this function in *C. militaris* is not yet clear [96,97,98]. The regulation of growth and development of *C. militaris* is a complex process involving multiple genes and metabolic networks, so the regulation mechanism of growth and development needs to be further explored.

**Table 2 bioengineering-11-00783-t002:** A list of genes that regulate the development of mycelium and fruiting body in *Cordyceps militaris*.

Gene Name	Accession Number	Annotation	Function	Reference
Cry-DASH	CCM_00774	Cryptochrome DASH	Fruiting-body development regulation	[95]
WC-1	AGO64764	Blue light receptor white collar 1	Mycelium growth and fruiting-body development regulation	[78]
VVD	CCM_04514	GATA transcription factor LreA	Fruiting-body development regulation	[79]
Lec3	CCM_01589	Lectin family integral membrane protein, putative	Fruiting-body development regulation	[94]
Lec4	CCM_03832	Ricin B-related lectin	Fruiting-body development regulation	[38]
Hyd1	CCM_03537	Hydrophobin 2	Fruiting-body development regulation	[84]
Hyd4	CCM_07964	Hydrophobin	Fruiting-body development regulation	[85]
fhp	CCM_05119	Flavohemoprotein	Fruiting-body development regulation	[99]
Snf1	CCM_05552	Carbon catabolite derepressing protein kinase	Fruiting-body development regulation	[100]
Chi1	CCM_08231	Class V chitinase, putative	Fruiting-body development regulation	[101]
Chi4	CCM_04817	Class V chitinase	Fruiting-body development regulation	[101]
crf1	CCM_07998	Fungal-specific transcription factor	Fruiting-body development regulation	[102]

## 5. Breeding for Disease Resistance of *C. militaris* Using CRISPR/Cas9

The “white mildew” disease caused by *Calcarisporium cordycipiticola* significantly affects the quality and yield of *C. militaris* in industrial production [25], which invades *C. militaris* through the gaps in the fruiting body mycelium without forming specialized infection structures, leading to cell wall dissolution, cell deformation, loss of organelles, and ultimately cell rupture [103]. Further analysis revealed that iron is an important virulence factor for the pathogen, and the genes Cmhsp78, Cmhsp70, and Cmhyd1 in *C. militaris* respond rapidly during the infection process and could serve as potential resistance genes for breeding disease-resistant strains of *C. militaris* [26]. Building on this foundation, researchers have utilized CRISPR/Cas9 gene-editing technology to overexpress the resistance gene Cmhyd1, obtaining a *C. militaris* strain with significantly enhanced resistance to this disease; this is the first disease-resistant gene-editing breeding system in mushrooms [27].

Fungi have specific temperature requirements during the growth phase, and temperatures that are too high or too low affect them. For instance, the fruiting-body growth of *Agaricus bisporus* is severely reduced at high temperatures of 33 °C, with a notable reduction in cap diameter and biomass [104]. The growth and hyphal morphology of *Tuber borchii* are affected at temperatures of 28 °C or 34 °C [105], while 42 °C inhibits the growth of *Ganoderma lucidum* mycelium, reduces branching, and induces the accumulation of ganoderic acid within the mycelium. The optimum growth temperature of *C. militaris* is 20–22 °C, and exposure to 25 °C during the later stages of growth can increase the content of cordycepin and carotenoids, whereas mycelial growth is severely inhibited at 30 °C [106]; therefore, the construction of heat-resistant strains is of great significance to the development of *C. militaris* industry. However, the specific temperature-stress response mechanisms in *C. militaris* remain unclear. Studies have suggested that introducing small heat shock protein genes into organisms can enhance their thermotolerance [107,108,109], which provide a reference for studying the heat stress response mechanisms in *C. militaris*.

## 6. Resource Utilization of Agricultural Waste and Sustainable Production of *C. militaris*

Agricultural waste refers to the organic substances discarded during the entire process of agricultural production, including the waste generated in agricultural production, livestock, and poultry farming. The spent mushroom substrate (SMS) is the residual biomass generated after harvesting the fruit bodies of edible/medicinal fungi, which are rich in lignocellulose, protein, and other essential elements [110,111]. They are generally disposed of by landfilling, open burning, and composting with animal manure, which are prone to causing environmental issues such as soil contamination, and air and water pollution [112,113]. Hence, more effective and efficient utilization of SMS is in demand for environmental protection, waste recycling, and sustainable development of agricultural resources and environment [114,115]. Although SMS has been used to produce industrial products such as packaging and building materials, biofuels, chemicals, and enzymes [116,117], these traditional recycling methods are time-consuming, with low conversion rates and added value.

*C. militaris*, an edible and medicinal fungus, contains an active substance pentostatin approved by the FDA as a first-line drug for the treatment of leukemia. Currently, pentostatin is produced by large-scale fermentation of *Streptomyces antibioticus* [118], but the yield is low. Since *C. militaris* lacks lignocellulose-degrading enzymes, the researchers introduced an exogenous optimized cellulase system, including XynA (xylanase from *Bacillus* sp. *KW*1), TrCBH1 (cellobiohydrolase from *Trichoderma reesei*), and CtCBH1 (cellobiohydrolase from the thermophilic fungus *Chaetomium thermophilum*), that can effectively degrade spent mushroom substrates (SMS) into glucose and other simple fermented sugars, thus enabling *C. militaris* to utilize the abundant lignocellulose produce the high-value anticancer drug pentostatin (Figure 4). Compared to the wildtype, the content of pentostatin in the engineered fungus significantly increased from 23.75% (CmctC) to 96.20% (CmTX). The big increase in output could be attributed to the multifunctional XynA and high-expressed TrCBH1, and the combined action of these enzymes could potentially enhance the capacity for hydrolyzing the fungal cell wall. Consequently, the destruction of the cell wall could stimulate the synthesis of secondary metabolites, akin to the stress response. There is a precursor competition relationship between the biosynthesis of pentostatin and cordycepin. To shift the metabolic flow towards the production of pentostatin and away from the rival biosynthesis pathway of cordycepin, the cns1 gene involved in cordycepin biosynthesis was removed in CmTX mutant, leading to pentostatin titers exceeding 600 mg/L in submerged fermentation [82]. This research is the first to turn edible fungi into cell factories for anticancer drugs and successfully transform agricultural waste into high-value-added anticancer drugs.

## 7. Conclusions and Expectation

*Cordyceps militaris*, a fungus renowned for its diverse bioactive compounds and its significance in both culinary and medicinal practices, continues to attract significant attention in the biotechnological and pharmaceutical industries. This article offers a comprehensive overview of the progress in genomic editing techniques, including ATMT, PEG, and CRISPR/Cas9. The application of these techniques in *Cordyceps militaris* is also summarized, including regulating key genes in the synthesis pathway of secondary metabolites to increase yield, regulating genes affecting growth and development to promote growth, reducing culture costs, and enhancing the expression of disease-resistant genes to obtain disease-resistant strains. The development of pathogen-resistant strains of *C. militaris* is an area ripe for exploration, as it would significantly mitigate the risks associated with cultivation and ensure sustainable production. Additionally, through heterologous expression of the cellulase gene, *Cordyceps militaris* can degrade agricultural wastes to produce high-value-added drugs. This sustainable production method is crucial for achieving a balance between environmental protection and economic growth, which would not only reduce the strain on natural resources but would also present a viable solution for large-scale cultivation. Moreover, the fusion antimicrobial peptide gene of Magainin II-Cecropin B was successfully expressed in *C. militaris*, and the resulting recombinant antimicrobial peptide, with its broad-spectrum antibacterial properties, particularly against *Staphylococcus aureus*, highlights the potential of *C. militaris* as a versatile platform for the production of therapeutic and functional biomolecules [35]. In the future, the combination of gene editing and synthetic biology holds immense potential for the creation of novel *C. militaris* strains capable of producing an expanded array of bioactive substances. Furthermore, the integration of high-throughput sequencing and omics technologies will undoubtedly deepen our knowledge of the *C. militaris* genome and transcriptome and speed up the discovery of novel genes and regulatory networks, thereby providing a broader palette of genetic modifications to enhance the production yield and efficiency of bioactive components. In conclusion, these advancements are poised to cater to the increasing demands of the pharmaceutical, health, and industrial sectors. Continued research on and development of *C. militaris* will undoubtedly lead to groundbreaking discoveries and applications in various fields, further solidifying its status as a valuable resource in the biomedical and industrial sectors.

## Figures and Tables

**Figure 1 bioengineering-11-00783-f001:**
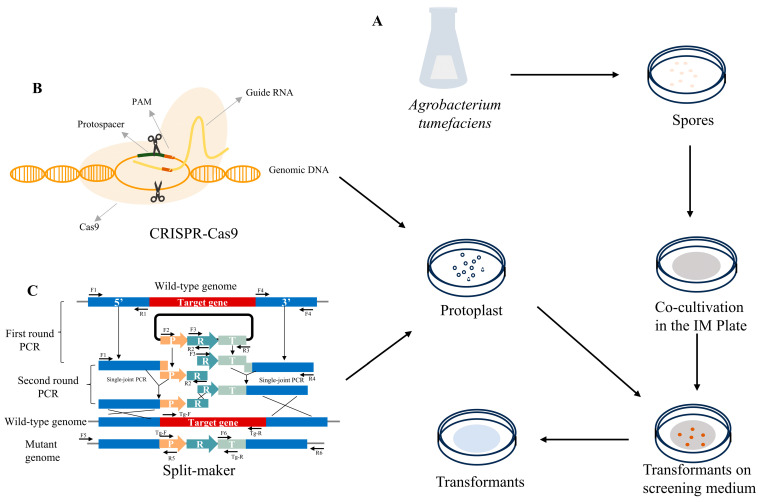
Genetic manipulation technologies of *C. militaris*. (**A**) *Agrobacterium*-mediated transformation (ATMT) process; the plasmid was introduced into *Agrobacterium* and cocultivated with *C. militaris* spores in the IM plate (induction solid medium used for cocultivation), then the transformants (mutants) were obtained. (**B**) CRISPR-Cas9 editing technology. The plasmid constructed by CRISPR-Cas9 was introduced into *C. militaris* protoplasts through the protoplast-mediated transformation (PMT) method, then the transformants were screened on corresponding resistance plates. (**C**) Split-marker approach. Split-marker deletion cassettes constructed using single-joint PCR (SJ-PCR) and double-joint PCR (DJ-PCR) were introduced into *C. militaris* protoplasts, then the transformants were screened on corresponding resistance plates. All transformants (mutans) were confirmed by using polymerase chain reaction (PCR) or the detection of a reporter gene.

**Figure 2 bioengineering-11-00783-f002:**
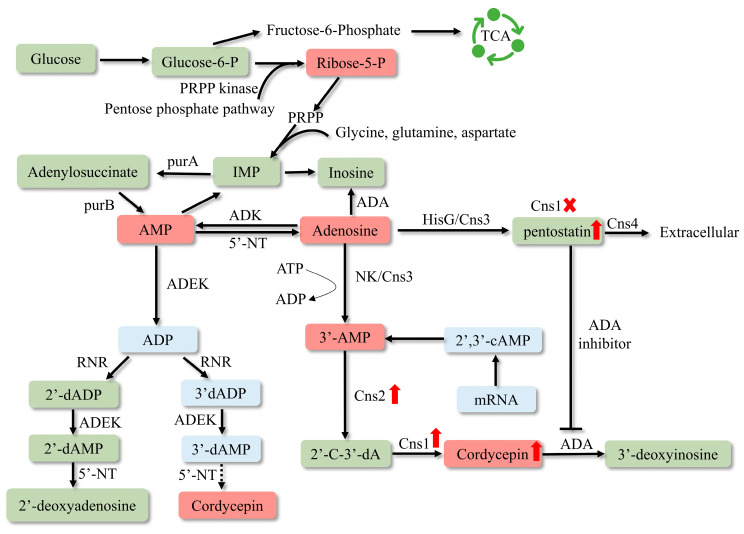
Biosynthetic pathway of cordycepin. PRPP—5-phosphoribosyl 1-pyrophosphate; TCA—tricarboxylic acid cycle; IMP—inosine monophosphate; RNR—ribonucleotide reductases; ADEK—adenylate kinase; NT5E—5′-nucleotidase; 2′-C-3′-dA—2′-carbonyl-3′-deoxyadenosine; NK—N-terminal nucleoside kinase; HisG—C-terminal HisG family of ATP phosphoribosyltransferases. The deletion of cns1 causes the metabolic flow to be inclined towards pentostatin synthesis, thus pentostatin content has been enhanced.

**Figure 3 bioengineering-11-00783-f003:**
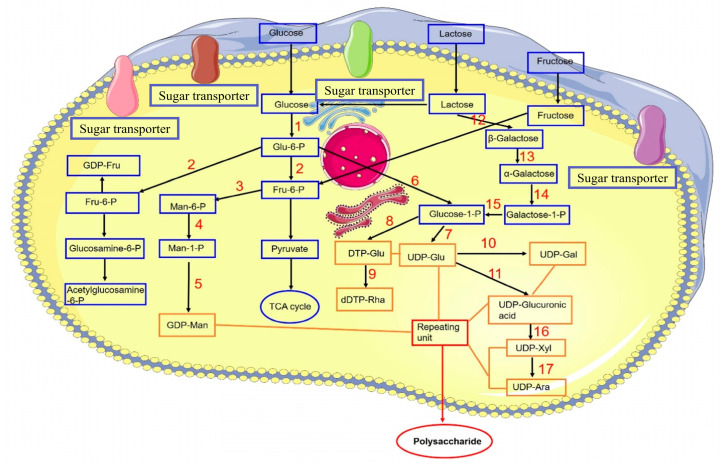
Proposed biosynthetic pathway of polysaccharides from *C. militaris* [55]. 1. Glucokinase. 2. Phosphoglucose isomerase. 3. Phosphate mannose isomerase. 4. Phosphomannose mutase. 5. GDP-mannose pyrophosphorylase. 6. α-Phosphoglucose mutase. 7. UDP glucose pyrophosphorylase. 8. dTDP-glucose pyrophosphorylase. 9. Rhamnose synthase. 10. UDP-glucose-4-epimerase. 11. UDP-glucose dehydrogenase. 12. β-Galactosidase. 13. Galactosidase. 14. Galactokinase. 15. UDP-glucose–hexose-1-phosphate uridylyltransferase. 16. UDP-xylose synthase. 17. UDP-xylose-4-epimerase.

**Figure 4 bioengineering-11-00783-f004:**
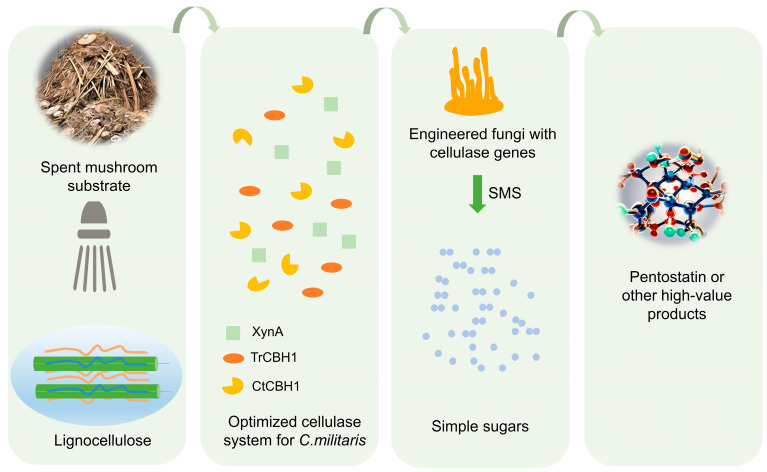
Engineered *C. militaris* with cellulase genes convert spent mushroom substrates (SMS) to valuable products. XynA—xylanase from *Bacillus* sp. *KW*1; TrCBH1—cellobiohydrolase from *Trichoderma reesei*; CtCBH1—cellobiohydrolase from the thermophilic fungus *Chaetomium thermophilum*. The optimized cellulase system was introduced into *C. militaris*, which was able to degrade SMS into monosaccharides, thereby promoting its own growth and increasing the content of the target product.

**Table 1 bioengineering-11-00783-t001:** Genetic manipulation technologies of *C. militaris*.

Strains	Method	Selection Marker	Target Gene	Transformation Efficiency	References
JM4	*Agrobacterium tumefaciens*-mediated transformation (ATMT)	Hygromycin B	-	30–600 cfu/ 1 × 10^5^ spores	[28]
HF 374-1, HF 432-1, HF 432-2, HF 438, and CM 001-5	Protoplast-mediated transformation (PMT)	Benomyl	*LaeA* (global regulator)	7 cfu/μg	[41]
CM10	Split-Marker	Basta	*Tns* (encoding a terpenoid synthase)	4.53 cfu/μg	[33]
CM10	CRISPR-Cas9	Basta	*pyrG* (encoding orotic acid-5′-monophosphate decarboxylase)	1.7 cfu/μg	[14]
CM15	CRISPR-Cas9	Basta	*Cns*1 (oxidoreductase domain-containing protein)	5 cfu/μg	[40]
CGMCC 3.16323	CRISPR-Cas9	hygromycin	*Cmwc*-1(blue light receptor white collar 1)	5.5 cfu/μg	[16]
			*Cmvvd* (GATA transcription factor LreA)	8.8 cfu/μg	
CGMCC 3.16323	CRISPR-Cas9	hygromycin	*Hyd*1 (Hydrophobin 2)	-	[27]

## Data Availability

Data are contained within the article.

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
