# Peer review of "Advancing Cordyceps militaris Industry: Gene Manipulation and Sustainable Biotechnological Strategies"

_bioengineering, 2024, doi:10.3390/bioengineering11080783_

Round 1

Reviewer 1 Report

Comments and Suggestions for Authors

In this article, Hu et al have reviewed the genomic editing techniques in C. militaris and their applications to improve desired traits in C. militaris such as increasing cordycepin production, improving polysaccharide production, other secondary metabolites, growth and development, and disease resistance. The authors also suggest engineering of C. militaris for utilization of residues from agriculture and mushroom cultivation. The authors have included relevant tables and figures where necessary. Overall, the review is well written and covers the area quite well. The authors should address the points below to address the shortcomings of the article:

Major comments

1.       The abstract needs improvement- it should cover the various products that have been improved through genetic engineering (as covered in the article).

2.       There are several spelling and formatting mistakes throughout the article. These must be checked and corrected. Some examples are given below:

a.       Line 78: protoplast need not start with a capital letter.

b.       Section 2.2: There are several instances of usage of “maker” instead of “marker”. They all should be corrected.

c.       The term “°C” seems to be written with different font types throughout the text.

3.       Lines 348 – 351 are not clear. Why would expression of biomass degrading enzymes increase pentostatin. What engineering were done to increase pentostatin by 23.75% to 96.20%? Lastly, in the next line, the titers of 600 mg/L suggests that it was a liquid culture. These lines should be rewritten to correctly represent the research that is cited.

Comments on the Quality of English Language

English quality is satisfactory. Some edits are needed as provided in the comments.

Reviewer 2 Report

Comments and Suggestions for Authors

In this review the authors discuss the importance of Cordyceps militaris as a production strain for bioactive compounds. They discussed the genetic tools used for the genetic manipulation of the strain and the level of success and reliability of the tool.

They also describe several cases where the strain is manipulated to overproduce valuable compounds. They also explored the diseases impacting the production of  the fungus.

In conclusion, the review covers different angles to make a Cordyceps militaris  a potential industrial strain.

My recommendation is to add another table that contains the compounds produced and the strategies used, and the titers obtained.

Reviewer 3 Report

Comments and Suggestions for Authors

The review manuscript by Hu and colleagues describes the potential of Cordyceps militaris for industrial exploitation in light of the recent development of genetic engineering and systems biology tools. The authors describe the different possibilities of genetic manipulation of the fungus as well as  the production a few substances with bioactive properties and ways to improve growth and cultivation of the fungus in general.

While I think the topic of the review itself is interesting and worth publishing, the manuscript is not yet in a mature shape. Also it needs careful revision of English language, esp. grammar and formal requirements. The text should be streamlined and the many different points below should be improved.

The main issues are:

-          English language, e.g. sentence in lines 8-10 is not correct, lines 54-57 are not conclusive, line 67/68 not correct, etc. Frequent spell errors, e.g. maker instead of marker, etc.

-          Figures 1,3 and 4  have not been mentioned /referenced in the text

-          Introduction does not mention the relation fo the fungus to insects, this is only mentioned very late in the text and thus confusing. The introduction should contain a few sentences on the biology of the fungus

-          Where is the genome sequence of the fungus deposited? Is there a paper on that?

-          Abbreviations are used inconsistently. Abbr. are also used without explanation in the abstract. Introduce abbr. and use consistently

-          A table to compare the three different methods of genetic engineering would be helpful along with the text

-          In part 2 the strategy of split marker system is not explained but shown in the figure. The authors should add primer binding sites in the figure and a brief explanation of the method in the text

-          Figure 1 should comprise numbers or a,b,c to explain the different paths and the text should relate to that by citing the figure and the fitting numbers. The plate labelled “transformants” is not clear for me, should this be the final mutants? What is IM (mentioned in figure but not in text). Do researchers also use the gold standard Southern blot for C.m?

-          Section 3: maybe resort the text and explain first COR, then ergothioneine as the ergoth. Section refers back to COR… or maybe this can even be combined as bioactive small molecules. Lines 238-248 are not really fitting to the rest in the ergothion. Part and is hard to follow. Polysaccharides should come after.

-          Mention what kind of molecule COR and ergoth. Are. The reader may not know.

-          Figure 2 is rather confusion and hard to grasp as the text does not really clearly describe the different known routes. Can you indicate in the figure the points that have been manipulated to enhance the biosynthesis? What means extracellular? Is this pentostatin?  Isn´t COR also extracellular and should you not mention that also?

-          Figure 3: What are the proteins in the membrane telling us (unlabelled?). Abbreviations are not well explained. Abbr. are also not explained in lines 221/222

-          Why is pentostatin not mentioned in the section on bioactive ingredients as it is discussed lateron and par tof the shown pathway?

-          Figure 4: the second panel is not clear to me. Are these enzymes (CAZymes) and biomass polymers (light blue dots)? It would be better if these polymers match the ones in the first panel or at least show some similar color or so.

Comments on the Quality of English Language

English language should be improved in terms of grammar and word spelling

Round 2

Reviewer 3 Report

Comments and Suggestions for Authors

The manuscript has significantly been improved and is fine for publication now.